# Deciphering the genetic landscape of tomato leaf curl New Delhi virus: Dynamic and region-specific diversity revealed by comprehensive sequence analyses

Zafar Iqbal ⬤*, Adil Alshoaibi, Sallah Ahmad Al Hashedi⬤, Muhammad Naeem Sattar, Khaled Muhammad Amin Ramadan⬤

Central Laboratories, King Faisal University, Al-Ahsa, Saudi Arabia

* zafar@kfu.edu.sa, zafariqbal2009@gmail.com

## Abstract

Tomato leaf curl New Delhi virus (ToLCNDV), a swiftly spreading bipartite begomovirus infecting ~43 plant species across Asia, Africa, and Europe, poses a major agricultural threat. This study comprehensively investigated the genetic diversity and evolutionary dynamics of ToLCNDV using a large dataset of sequences from diverse geographic regions across Asia, Africa, and Europe. Phylogenetic analysis of DNA-A (TA) and DNA-B (TB) components revealed seven major clades for each component, demonstrating regional but not host-specific clustering. Notably, TA and TB exhibited distinct geographic segregation and genetic diversity, with cognate segments often clustering separately. Extensive recombination events were detected (44 in TA and 45 in TB), involving both intra- and interspecies events. While TA displayed higher overall genetic diversity indices, TB exhibited greater nucleotide diversity ($\pi = 0.11\%$) compared to TA ($\pi = 0.057\%$) and a slightly faster evolutionary rate ($8.15 \times 10^{-04}$ substitutions/site/year vs. $7.25 \times 10^{-04}$ for TA). Demographic analysis indicated strong negative selection on both components (TA and TB) and their coding regions, albeit with varying intensity. India and Pakistan, known ToLCNDV hotspots, harbored the highest viral diversity and isolate numbers. This study highlights the complex interplay of genetic diversity, recombination, and selection in shaping ToLCNDV evolution, revealing distinct evolutionary trajectories for its genomic components and in different countries. These findings enhance understanding of ToLCNDV evolution and support informed control strategies.

## 1. Introduction

Tomato leaf curl New Delhi virus (ToLCNDV) is a most prevalent and widespread bipartite begomovirus (Family *Geminiviridae*) that occurs across three continents: Asia, Africa, and Europe. It is a major impediment to the production of many

which permits unrestricted use, distribution, and reproduction in any medium, provided the original author and source are credited.

**Data availability statement:** This article and its supplementary information files contain all data generated or analysed during this study. The accession numbers of the used ToLCNDV sequences are available in S1 Table and can be accessed directly by using accession number at NCBI GenBank (https://blast.ncbi.nlm.nih.gov/).

**Funding:** This scientific paper is "derived from a research grant funded by the Research, Development, and Innovation " Authority (RDIA) - Kingdom of Saudi Arabia - with grant number (12877-KFU-2023-KFU-R-2-1-SE-). While the publication of this research was supported by the Deanship of Scientific Research (DSR), King Faisal University, Kingdom of Saudi Arabia, (KFU252244). There was no additional external funding received for this study.

**Competing interests:** No competing interest among authors.

economically important crops, particularly tomatoes, in South and Southeast Asia [1,2]. ToLCNDV has a wide host range; besides tomatoes, it infects at least 43 other dicot plant species, including economically important vegetables, ornamental plant species, and weeds [3–14]. Although tomato leaf curl disease, which was thought to be caused by ToLCNDV, could be traced back to 1948 in India (Vasudeva and Raj, 1948), nonetheless, its first presence at molecular level was deciphered in 1995 [1]. Since its first revelation in 1995, it spread to Pakistan in 2004 [14], Bangladesh in 2005 [13], Iran in 2008 [15], Taiwan in 2010 [16], Thailand and Indonesia in 2010 [2], and Malaysia in 2016 [17]. ToLCNDV reached Europe in 2012 (Spain) [18] and later spread to Sicily and southern Italy in 2015 [6], Greece in 2018 [19], Portugal in 2019, and France in 2020 [20]. Similarly, ToLCNDV was identified in Africa in 2015 (Tunisia) [7], Morocco in 2017 [21], and China in 2022 [22].

ToLCNDV is a typical bipartite begomovirus with two circular and single-stranded (ss) DNA genome components, DNA-A (TA) and DNA-B (TB). These components encode eight genes on both orientation (Fig 1). The TA encodes proteins essential for encapsidation (coat protein, AV1/CP) and its precursor (pre-coat protein, AV2), along with other proteins involved in replication (AC1/Rep), regulating transcription (AC2/TrAP), and enhancing replication (AC3/REn). An additional protein, AC4, is encoded on TA in the virion-sense. The TB encodes two movement proteins, virion-sense protein is nuclear shuttle protein (NSP [BV1]), and complementary-sense protein is movement protein (MP [BC1]). The activities, structures and functions of these proteins have been thoroughly studied [23,24]. Like all geminiviruses, genes on ToLCNDV genome are separated by a non-coding intergenic region (IR), containing a common region (CR). The CR has a hairpin structure with a specific nonanulceotide sequence (TAATATT/AC) that marks the origin of replication [1]. Additionally, the CR contains short, repeating sequences (iterons) to which Rep binds to induce nick to commence replication. ToLCNDV requires both components for symptomatic infection; however, its TA alone can infect plants asymptomatically, albeit at a reduced rate after *Agrobacterium* inoculation [1,25]. ToLCNDV has been found associated with betasatellites under natural conditions in Pakistan and India [26–29]; additionally, a betasatellite impaired its ability to maintain cotton leaf curl Multan alphasatellite (CLCuMuA) under controlled conditions [30].

ToLCNDV is transmitted between plants by whiteflies, specifically *Bemisia tabaci* (Gennadius), with different cryptic species acting as vectors in distinct geographical regions. In Asia, the Asia II-5 cryptic species is the primary vector [31,32]. In Europe, vector diversity is higher, with the Mediterranean-Q1 cryptic species reported in Spain [33] and the MED-Q2 cryptic species in Italy [34]. In Africa (Seychelles), the Asia 1 cryptic species has been implicated [35]. Notably, not all whitefly species are competent vectors; *Trialeurodes vaporariorum* is non-transmissive, and the *B. tabaci* Mediterranean (Q) cryptic species shows low transmission efficiency between zucchini and the wild cucurbit, *Ecballium elaterium*, in Italy [36].

Viruses possess high mutation rates, large population sizes, and rapid replicative kinetics, contributing to the generation of a vast pool of viral variants [37]. Such closely related genomic variants (viral populations) possess a remarkable ability to

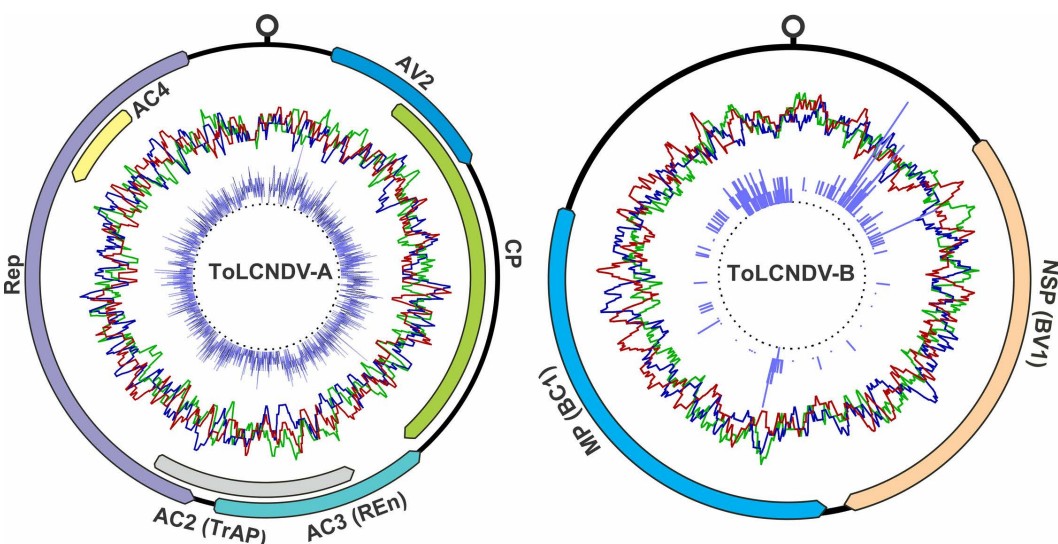

**Fig 1. The typical genome organization of ToLCNDV.** The outermost circle depicts the location and transcriptional orientation (arrows) of all open reading frames (ORFs) encoded on both DNA-A and DNA-B components. The origin of replication, characterized by a hairpin loop structure containing the conserved nonanucleotide sequence TAATATTAC, is also shown. The middle circle represents the distribution of GC and AT content across the genome, while the innermost circle displays the distribution of single nucleotide polymorphisms (SNPs) visualized by blue bars.

adapt and evolve, enabling them to overcome challenges and persist in diverse environments. In addition, single nucleotide polymorphisms (SNPs), caused by a single base change, in the genomes of viruses, occurring either in coding (non-synonymous) or non-coding regions (synonymous), can influence viral infection and gene expression [38]. Studying these mutations helps scientists track viral evolution and transmission because mutations in begomoviruses have been linked to increased spread [39,40].

RNA viruses, characterized by error-prone RNA-dependent RNA polymerases and consequently high mutation rates [41], have been the primary focus of phytovirus evolutionary studies and are generally considered to evolve rapidly than DNA viruses. However, ssDNA viruses, like ToLCNDV, pose a serious emerging threat to global agriculture and interestingly, their mutation rates and evolutionary imprints are akin to RNA viruses [42–44]. The rapid genetic diversification observed in ssDNA viruses is attributed to two main factors: low-fidelity DNA polymerase activity during replication and spontaneous biochemical reactions like methylation, deamination, and base oxidation [45,46]. While mutation dynamics are undeniably crucial in geminivirus diversification [47–49], they do not fully explain the observed genetic variation. Alongside point mutations and nucleotide substitutions, recombination events play a significant role in the evolution of geminiviruses [17,50,51]. ToLCNDV, a highly destructive plant pathogen, has swiftly evolved within diverse cropping systems and readily infects alternative hosts, even non-cultivated host reservoirs like weeds. Recurrent mutations, recombination, and the acquisition of satellite DNAs have presumably imparted a significant role to the emergence and success of ToLCNDV.

South Asia, a major tomato producer contributing over 30% of the global market (2.54 million tonnes annually; FAOSTAT 2022), accessed on April 03, 2024), has witnessed a concerning rise in begomoviruses spread, including ToLCNDV. This region is not only considered to be the origin of ToLCNDV, but also a hotspot for its diversification [3,17,22,52,53]. The high degree of ToLCNDV diversity in South Asia suggests a critical role for this region in shaping ToLCNDV global evolution and dissemination. Additionally, several new mutants and recombinant ToLCNDV variants have emerged, indicating a dramatic increase in ToLCNDV incidence [54,55].

While previous studies have comprehensively addressed various aspects of ToLCNDV [56,57], the information on its global evolutionary dynamics remains scarce. Additionally, there is a dearth of data concerning a comparative analysis of genetic variability and evolutionary aspects to comprehend ToLCNDV populations across diverse geographical locations. This study addresses these gaps, exploring the genetic diversity, recombination dynamics, and evolutionary trajectory of ToLCNDV. By focusing on genomic variability without a hypothesis-driven framework, this study aims to expand understanding of the factors shaping the virus's evolution and to provide a foundation for future research and management strategies.

## 2. Materials and methods

### 2.1 Sequence retrieval and multiple sequence alignment

To investigate the genetic diversity of ToLCNDV, only available full-length complete genome sequences were retrieved from NCBI GenBank (https://www.ncbi.nlm.nih.gov/) on January 22nd, 2024. These sequences were processed to create nucleotide datasets for the DNA-A (TA) and DNA-B (TB) components. From these datasets, sub-datasets were generated for each open reading frame (ORF). Additionally, sequences were categorized by country of origin, with further division into their respective ORFs to facilitate country-specific analyses. Multiple sequence alignments (MSA) were then performed on all the generated datasets using Muscle within MEGA11 software [58,59]. These alignments were meticulously reviewed and manually adjusted to ensure accurate sequence alignment.

### 2.2 Evolutionary time estimation, phylogeny, and SNPs

The generated MSA files of full-length TA and TB datasets were used for phylogenetic inference. For phylogeny, the best fit nucleotide substitution model was determined using MEGA11, a user-friendly platform supporting diverse evolutionary models with simplifies phylogenetic analyses. Subsequently, phylogenies were inferred using the maximum-Likelihood (ML) method and each analysis employed 1000 bootstrap replicates to assess nodal support. The resulting ML trees of TA and TB were visualised and edited in Interactive Tree of Life (iTOL v6.5; https://itol.embl.de/#), accessed on June 2024) [60].

### 2.3 ToLCNDV population structure assay

DNA polymorphism analysis was performed using DnaSP v.6.12 [61], a crucial program for population genetics studies, calculating genetic diversity measures and detecting selection signatures in large datasets. Nucleotide diversity ($\pi$), a measure of average nucleotide differences per site within a population, was computed for all datasets (TA, TB, and all ORFs). Details information regarding all the inferred nucleotide diversity attributes has been previously outlined [48]. Briefly, significant differences in mean $\pi$ values across all datasets were assessed using 95% bootstrap confidence intervals with a window size of 100 nucleotides and a step size of 25 nucleotides. This approach allows for the identification of regions with significant variation within the sequences.

DnaSP was further employed to conduct neutrality tests, including Tajima's D (TD) and Fu and Li's D (FLD), across all datasets. These tests aimed to assess genetic diversity, focusing on parameters such as the number of segregating sites and the mean number of pairwise differences. DnaSP software was further utilized to identify single nucleotide polymorphisms (SNPs) across the entire genomes of ToLCNDV populations (TA and TB) on a global scale. Additionally, the distribution of GC and AT content across all TA (662 sequences) and TB (535 sequences) genomes were inferred in Geneious program (https://www.geneious.com/).

### 2.4 Estimation of nucleotide substitution rate

Nucleotide substitution rates (substitutions per site per year) within the TA, TB and their ORFs was estimated using a Bayesian Markov chain Monte Carlo (MCMC) approach implemented in BEAST (v.1.10.5) [62]. BEAST's flexibility allows for complex evolutionary modelling, making it ideal for studies needing precise timescale and rate estimates. Each dataset was analysed independently using relaxed molecular clocks with uncorrelated lognormal distribution to estimate mutation

rates at three different codon positions. The MCMC chains were run for a length of $1 \times 10^8$ to ensure an effective sample size of ≥200 for all parameters.

### 2.5 Estimation of selection pressure

To assess selection pressures acting on ToLCNDV-encoded ORFs, the ratio of non-synonymous to synonymous substitutions (dN/dS) was calculated using two distinct approaches. MEGA11, employing the Nei-Gojobori method (*p*-distance), was used for initial dN/dS estimations. To provide a more robust analysis, Datamonkey, an online webserver [63], was utilized with two independent methods: Fast, Unconstrained Bayesian AppRoximation (FUBAR) and Fixed Effects Likelihood (FEL). Furthermore, DnaSP was employed to evaluate selection pressure at the population level.

### 2.6 Recombination analysis

To identify potential recombination events within the TA and TB genomes, we employed two established methods: the Genetic Algorithm for Recombination Detection (GARD) (https://www.datamonkey.org/gard) and the Recombination Detection Program (RDP v.5.5) [64]. For RDP analysis, we implemented seven algorithms (BOOTSCAN, CHIMAERA, GENECONV, RDP, MAXCHI, SISCAN, and 3SEQ) with default detection thresholds and a Bonferroni-corrected *p*-value of 0.05 to infer recombination breakpoints with high confidence. Events identified by at least five algorithms were considered credible recombination events. To corroborate the RDP findings, we performed GARD analysis using the Beta-Gamma site-to-site variation model, four rate classes, and normal run mode.

## 3. Results

### 3.1 Geographical distribution of ToLCNDV

A total of 662 TA and 535 TB full-length sequences (Tables 1 and S1, Fig 2) were analyzed, spanning 43 plant species (such as tomato, potato, chili, cucurbits, melons, zucchini, gourds, and weeds) across 17 Asian, African, and European

Table 1. Total number of TA and TB sequences from various countries included in the nucleotide diversity analysis.

| Country | No. of sequences | |
|---|---|---|
| | TA | TB |
| Algeria | 1 | 0 |
| Bangladesh | 12 | 4 |
| Cambodia | 19 | 0 |
| China | 27 | 27 |
| France | 1 | 0 |
| India | 122 | 94 |
| Indonesia | 3 | 3 |
| Iran | 6 | 7 |
| Italy | 5 | 5 |
| Laos | 3 | 0 |
| Morocco | 1 | 0 |
| Pakistan | 352 | 279 |
| Seychelles | 1 | 0 |
| Spain | 93 | 104 |
| Taiwan | 1 | 0 |
| Thailand | 7 | 4 |
| Tunisia | 8 | 8 |
| **Total** | **662** | **535** |

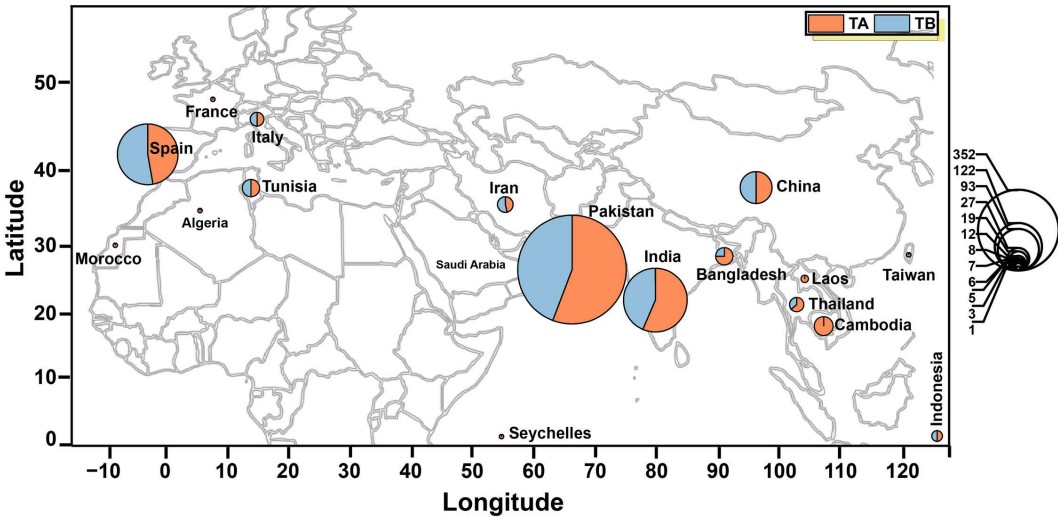

**Fig 2. Geographical distribution and population density map of ToLCNDV DNA-A (TA shown in brown) and ToLCNDV DNA-B (TB shown in blue) from the Old World.** The used map was taken from NASA's public database (https://eol.jsc.nasa.gov/) and is available for use without copyright permission.

countries. The majority of TA sequences were from Pakistan (352), followed by India (122), Spain (93), China (27), Cambodia (19), Bangladesh (12), and Tunisia (8). Likewise, most TB sequences were reported from Pakistan (279), followed by Spain (104), India (94), China (27), Tunisia (8), Iran (7), Italy (5), Bangladesh (4), Thailand (4), and Indonesia (3). Notably, aside from the presence of TA, no full-length TB sequence had been reported from Algeria, Cambodia, France, Laos, Morocco, Seychelles, and Taiwan up to the commencement of this study.

ToLCNDV exhibits a broad host range with distinct geographic patterns across three continents. In Asia, the virus primarily infects tomato (India, Pakistan, Bangladesh, Thailand, Iran), cucurbits (pumpkin, cucumber, bottle gourd, and bitter gourd in India, Thailand, Indonesia), potato (India, Pakistan), okra (India), cotton (Pakistan), papaya (Bangladesh, India), and chilli (India). Europe reports infections in zucchini, melon, and tomato (Spain, Italy), with recent emergence in pumpkin (France). In Africa, ToLCNDV affects cucumber, melon, and zucchini (Tunisia, Morocco, Algeria). Additionally, the virus infects weeds (bindweed in Pakistan) and ornamental plants (chrysanthemum in India), demonstrating its adaptability to diverse plant species.

### 3.2 Phylogenetic analysis

The initial ML phylogenetic analysis of full-length 662 TA and 535 TB sequences with inferred best fit model (GTR + G for both TA and TB) revealed 12 and 17 major clades, respectively, indicating regional groupings without distinct host-specific demarcations. To reduce redundancy, we excluded identical sequences from the same location and host, resulting in 219 TA and 161 TB sequences. Subsequent phylogenetic analysis of these reduced datasets identified the same number of major clades in TA and TB, further supporting the dominance of regional clustering with limited evidence of host-specific segregation (Fig 3). Phylogenetic analysis reveals distinct, geographically clustered clades of ToLCNDV isolates from various regions, including Spain, Tunisia, Morocco, India, and Pakistan. The presence of closely related isolates across regions suggests recombination or recent introductions, while genetic divergence within some isolates likely reflects evolutionary pressures or host adaptation. Among these, the oldest, most recent and most divergent ToLCNDV population was traced back to India. Notably, close relationships were observed between some Indian and Spanish isolates in both TA

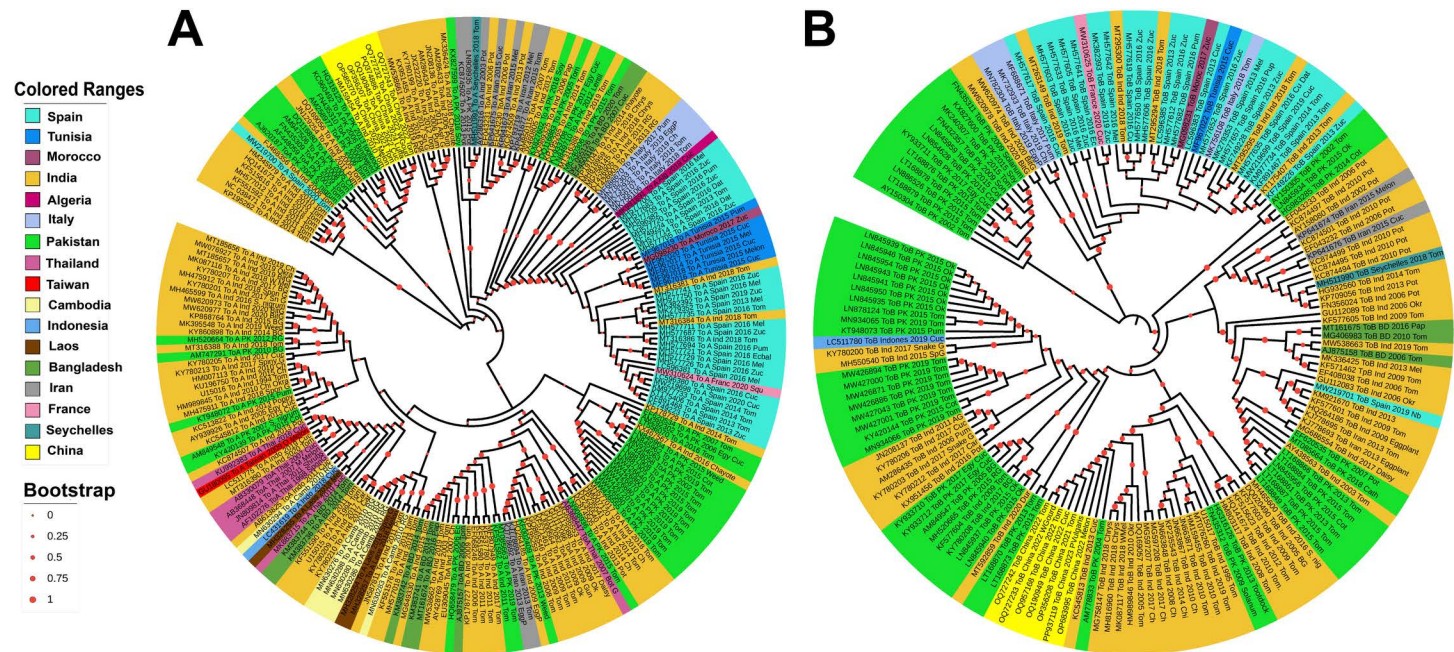

**Fig 3. The phylogenetic dendrogram of selected sequences of ToLCNDV, TA (A) and TB (B), based on their respective full-length sequences.**
The phylogenetic tree was constructed with Maximum-Likelihood (ML) algorithm in MEGA11 with 1000 bootstrap values. Geographic origin (country) of the isolates is indicated by the colored circle.

(MT316381, MT316384, MT316386, and MW219700) and TB (e.g., MT295294, MT295295, MT295300, KT175307, and MW219701), with isolates from each region clustering together.

### 3.3 Recombination analysis

**3.3.1 RDP.** Two approaches were used to detect potential recombination events in the ToLCNDV population. First, global datasets of TA-all (662 sequences) and TB-all (535 sequences) were analyzed (S2 Table). Second, datasets were analyzed by individual country (Table 2). Global dataset analysis revealed the presence of 98 potential recombination events across 68 TA isolates, with 44 deemed credible within hotspot nucleotide regions: 100–200, 600–625, 700–725, 1400–1425, 2000–2200, and 2333–2350 (S2 Table). Overall, the highest number of breakpoints were determined in Asian isolates, with one Indian isolate (MK336424) showing four, and five other isolates (JX460805, KF551592, KP195265, MH475912, and KY420139) showing three each. Only two Chinese isolates (OQ957166 and OQ190947) showed single recombination events. In TB isolates, 119 potential recombination events were found across 86 isolates, with 45 credible within hotspot regions: 500–550, 1200–1300, 2000–2300, and 2400–2600 (S2 Table). Similar to TA, Asian isolates showed the most recombination. Eight isolates (KF515623, KF577603, MG597209, MT592859, MT592862, MT592863, LN845940, and LT168873) had three recombination events each. It is important to mention that the majority of inferred recombination breakpoints were interspecies, indicating recombination between different ToLCNDV strains/ isolates. However, several intraspecies recombination events were also detected, demonstrating recombination with other begomovirus species.

Analysis of recombination events by country of origin (Table 2) revealed a strong concentration of potential recombination breakpoints in Asian isolates. Pakistani isolates exhibited the highest number, with 13 breakpoints in TA and 10 in TB sequences, followed by Indian isolates (11 in TA and 7 in TB), Bangladeshi isolates (9 in TA and 3 in TB), and Iranian

**Table 2. Recombination breakpoints in ToLCNDV TA and TB components across different countries.**

| Dataset | Virus | Number of breakpoints | Position | | Detection Methods[a] | *p*-value[b] |
|---------|-------|----------------------|----------|-----|---------------------|-------------|
| | | | start | end | | |
| Bangladesh | TA | 9 | 2489 | 156 | RBMC**S**3 | $1.95 \times 10^{-15}$ |
| | | | 2716 | 38 | RMC**S**3 | $1.95 \times 10^{-15}$ |
| | | | 2666 | 968 | RBMC**S**3 | $5.29 \times 10^{-03}$ |
| | | | 2489 | 2670 | RBMC**S**3 | $1.95 \times 10^{-15}$ |
| | | | 2461 | 967 | RBMC**S**3 | $5.29 \times 10^{-03}$ |
| | | | 346 | 2146 | RMC**3** | $7.27 \times 10^{-09}$ |
| | | | 2135 | 2670 | RBMC**S**3 | $1.95 \times 10^{-15}$ |
| | | | 1950 | 2218 | RBMC**S**3 | $4.94 \times 10^{-07}$ |
| Cambodia | TA | 8 | 11 | 900 | RMC**S**3 | $3.41 \times 10^{-29}$ |
| | | | 2297 | 2744 | RGM**S**3 | $2.65 \times 10^{-15}$ |
| | | | 1967 | 20 | RMC**S**3 | $3.41 \times 10^{-29}$ |
| | | | 2054 | 2165 | GMC**S**3 | $1.21 \times 10^{-09}$ |
| | | | 938 | 1428 | GBMC**S**3 | $3.25 \times 10^{-53}$ |
| | | | 612 | 2316 | RGBMC**S**3 | $3.60 \times 10^{-18}$ |
| | | | 1758 | 2343 | BMC**S**3 | $7.55 \times 10^{-29}$ |
| | | | 426 | 900 | GMC**S**3 | $3.25 \times 10^{-53}$ |
| China | TA | 1 | 2080 | 6 | **M**CS3 | $1.19 \times 10^{-08}$ |
| India | TA | 11 | 96 | 1969 | MC**S**3 | $7.83 \times 10^{-03}$ |
| | | | 1890 | 2144 | MC**S**3 | $8.86 \times 10^{-05}$ |
| | | | 2258 | 3 | RBMC**S**3 | $5.59 \times 10^{-03}$ |
| | | | 995 | 2069 | RGBMCS**3** | $2.08 \times 10^{-37}$ |
| | | | 1475 | 2169 | RBM**S**3 | $4.55 \times 10^{-21}$ |
| | | | 2277 | 2635 | RG**B**MCS3 | $9.41 \times 10^{-03}$ |
| | | | 1004 | 1731 | RMC**S**3 | $4.11 \times 10^{-08}$ |
| | | | 643 | 2050 | RGBMCS**3** | $8.30 \times 10^{-26}$ |
| | | | 1330 | 2310 | RGBMC**S**3 | $1.53 \times 10^{-23}$ |
| | | | 724 | 1441 | RGBMC**S**3 | $9.64 \times 10^{-63}$ |
| | | | 264 | 1043 | B**M**CS3 | $7.94 \times 10^{-07}$ |
| Iran | TA | 3 | 1435 | 2415 | MC**S**3 | $9.96 \times 10^{-27}$ |
| | | | 1910 | 46 | RBMC**S**3 | $7.94 \times 10^{-07}$ |
| | | | 1825 | 2610 | MC**S**3 | $9.96 \times 10^{-27}$ |
| Italy | TA | 2 | 2715 | 31 | BMC**S**3 | $3.28 \times 10^{-10}$ |
| | | | 818 | 1850 | **B**MCS3 | $2.33 \times 10^{-07}$ |
| Pakistan | TA | 13 | 1460 | 2692 | GBMC**3** | $1.45 \times 10^{-23}$ |
| | | | 854 | 1868 | RGBMC**S**3 | $1.53 \times 10^{-12}$ |
| | | | 2453 | 2739 | RMC**S**3 | $5.93 \times 10^{-10}$ |
| | | | 130 | 545 | GMC3 | $2.98 \times 10^{-05}$ |
| | | | 1027 | 2055 | RGMC**S**3 | $3.63 \times 10^{-10}$ |
| | | | 2416 | 184 | **R**GBMCS3 | $2.64 \times 10^{-15}$ |
| | | | 2179 | 705 | RGMC**S**3 | $3.63 \times 10^{-10}$ |
| | | | 1623 | 2375 | MC**S**3 | $8.06 \times 10^{-03}$ |
| | | | 526 | 2189 | **R**GBMCS3 | $3.13 \times 10^{-32}$ |
| | | | 190 | 1633 | GBMC**S** | $2.21 \times 10^{-19}$ |
| | | | 1149 | 1450 | RGMC**S**3 | $3.82 \times 10^{-08}$ |
| | | | 1142 | 1301 | RG**B**MCS3 | $1.49 \times 10^{-16}$ |
| | | | 1128 | 1454 | RGBMC**S**3 | $1.53 \times 10^{-12}$ |

*(Continued)*

**Table 2.** (Continued)

| Dataset | Virus | Number of breakpoints | Position | | Detection Methods[a] | p-value[b] |
|---------|-------|----------------------|----------|-----|--------------------|-----------|
| | | | start | end | | |
| Spain | TA | 0 | 0 | 0 | 0 | 0 |
| Thailand | TA | 4 | 2420 | 427 | R**M**C3 | $9.71 \times 10^{-03}$ |
| | | | 981 | 1917 | RM**C**3 | $6.44 \times 10^{-03}$ |
| | | | 37 | 2338 | RMC**S**3 | $6.56 \times 10^{-06}$ |
| | | | 910 | 1432 | RGM**S**3 | $6.48 \times 10^{-06}$ |
| Tunisia | TA | 2 | 90 | 593 | RG**M**3 | $4.89 \times 10^{-05}$ |
| | | | 912 | 1278 | B**M**S3 | $7.88 \times 10^{-05}$ |
| Total 58 | | | | | | |
| Bangladesh | TB | 3 | 257 | 1159 | RGBMC**S**3 | $7.10 \times 10^{-15}$ |
| | | | 590 | 1375 | RMC**S**3 | $6.40 \times 10^{-14}$ |
| | | | 1349 | 2189 | RGMC**S**3 | $2.52 \times 10^{-12}$ |
| China | TB | 1 | 2189 | 2389 | RMC**S** | $2.21 \times 10^{-03}$ |
| India | TB | 7 | 577 | 996 | RGBMC**S**3 | $8.37 \times 10^{-08}$ |
| | | | 263 | 568 | R**G**BMCS3 | $4.81 \times 10^{-11}$ |
| | | | 1325 | 2166 | RGMC**S**3 | $1.70 \times 10^{-14}$ |
| | | | 922 | 1371 | RMC**S**3 | $2.13 \times 10^{-15}$ |
| | | | 144 | 673 | RGBMC**S**3 | $2.91 \times 10^{-09}$ |
| | | | 2555 | 139 | RGBMC**S**3 | $2.69 \times 10^{-22}$ |
| | | | 1032 | 2151 | RGBMC**S**3 | $1.38 \times 10^{-44}$ |
| Iran | TB | 4 | 674 | 879 | **R**MCS3 | $7.17 \times 10^{-04}$ |
| | | | 2157 | 2631 | R**G**BMC3 | $6.97 \times 10^{-08}$ |
| | | | 1524 | 1964 | GBMC**S**3 | $1.05 \times 10^{-25}$ |
| | | | 2553 | 489 | GMC**S** | $1.11 \times 10^{-08}$ |
| Italy | TB | 1 | 2477 | 168 | MC**S**3 | $1.30 \times 10^{-03}$ |
| Pakistan | TB | 10 | 738 | 2337 | RGBMC**S**3 | $2.47 \times 10^{-12}$ |
| | | | 291 | 1241 | RGBMC**S**3 | $1.51 \times 10^{-16}$ |
| | | | 1053 | 2648 | RGBMC**S**3 | $2.47 \times 10^{-15}$ |
| | | | 1795 | 2679 | RGM**S**3 | $5.28 \times 10^{-16}$ |
| | | | 284 | 1180 | RGMC**S**3 | $8.16 \times 10^{-20}$ |
| | | | 263 | 543 | RGMC**S**3 | $6.48 \times 10^{-12}$ |
| | | | 1408 | 2174 | GBMC**S**3 | $1.91 \times 10^{-12}$ |
| | | | 659 | 960 | RGBMC**S**3 | $2.59 \times 10^{-10}$ |
| | | | 2182 | 2739 | RMC**S**3 | $3.49 \times 10^{-4}$ |
| | | | 629 | 2079 | RGBCM**S**3 | $1.01 \times 10^{-64}$ |
| Spain | TB | 1 | 355 | 2174 | GBMC**S**3 | $1.91 \times 10^{-12}$ |
| Thailand | TB | 2 | 423 | 1636 | MC**S**3 | $2.34 \times 10^{-10}$ |
| | | | 2167 | 2509 | **MCS3** | $1.31 \times 10^{-06}$ |
| Tunisia | | 1 | 1550 | 2378 | R**G**BMC3 | $4.81 \times 10^{-11}$ |

[a]B, Bootscan; C, Chimaera; G, GeneConv; L, LARD; M, MaxChi; P, Phylpro; R, RDP; S, SisScan; 3, 3SEQ.

[b]The lowest p-value corresponds to the recombination program (bold) is mentioned.

Abbreviation used in the table are tomato leaf curl New Delhi virus DNA-A (TA) and DNA-B (TB).

isolates (3 in TA and 4 in TB). Notably, Cambodian TA isolates also showed a high number of recombination breakpoints (8). European isolates showed considerably less recombination, with Italian isolates having 2 breakpoints in TA and 1 in TB, and Spanish isolates showing no breakpoints in TA and only 1 in TB. Among African isolates, Tunisian isolates had 2

breakpoints in TA and 1 in TB. This data highlights a geographical pattern in recombination frequency, with Asian isolates, particularly those from Pakistan and India, displaying the highest levels.

**3.3.2 GARD.** Recombination analysis using GARD, largely consistent with RDP results, identified numerous potential breakpoints in both TA and TB sequences. In TA, 1916 potential breakpoints were detected across 751 models, with 15 considered credible based on substantial improvements in corrected Akaike Information Criterion (ΔC-AIC) compared to both the null model (ΔC-AIC = 450.247) and the single-tree multiple-partition model (ΔC-AIC = 4731.31). These credible TA breakpoints clustered around nucleotide positions 100–200, 600–625, 700–725, 1000–1100, 1350–1400, 1600–1650, 2000–2200, and 2300–2400 (S1A Fig). Similarly, analysis of TB sequences revealed 2323 potential breakpoints from 1640 models, with 16 deemed credible (ΔC-AIC = 534.63 compared to the null model and ΔC-AIC = 5810.38 compared to the single-tree multiple-partition model). These credible TB breakpoints were primarily located around nucleotide positions 50, 200, 600–800, 1100, 1300, 1500, 1900–2150, 2500, 2650, and 2800 (S1B Fig).

Recombination breakpoint analysis (Fig 4) revealed a high prevalence of recombination, especially among Asian isolates. Pakistani and Indonesian isolates showed the most breakpoints (nine each in both TA and TB). In Pakistani isolates, breakpoints were located around nucleotide positions 200, 590, 1000-1100, 1350-1400, 1940, and 2450 (TA) and 300, 800–950, 1180, 1670, 2150, 2380, and 2780 (TB). Indonesian breakpoints occurred at 50, 90, 200, 280, 950, 1190, 1830-1850, and 2230 (TA) and 600, 1080, 1270, 1420, 1600–1800, 2150, and 2500 (TB), although these could not be reliably inferred by RDP due to a small sample size (n = 3). Among the other Asian countries, Indian and Thai isolates each had five breakpoints in both TA (India: 770, 1400–1500, 1950, 2325; Thailand: 420, 930, 1425–1450, 1910, 2290) and TB (India: 590, 800, 930, 1120, 2460, 2870; Thailand: 500–520, 875, 2090–2180, 2510). Bangladeshi isolates had six breakpoints in TA (450, 950, 1150, 1400, 1900, 2200) and nine in TB (20, 150, 190, 590, 1170, 1190, 1350, 2100, 2500). Iranian isolates had six breakpoints in TA (1430, 1480, 1620, 1790–1830, 2450–2500, 2610) and eleven in TB (450, 600–630, 900, 1490, 1990, 2150, 2210, 2300, 2590–2630, 2580). Cambodian TA isolates showed eight recombination breakpoints (410–500, 920, 1420, 1970, 2040). Consistent with RDP results, Chinese isolates showed only one breakpoint in each component: TA (2150) and TB (2390). European isolates showed least number of potential breakpoints: Italian isolates had 4 in TA (160, 825, and 1825, and 1950) and 2 in TB (150 and 2390), while Spanish isolates had only one breakpoint TB (2500) and none in TA. Among African isolates, Tunisian isolates had 3 breakpoints in TA (90, 9210, and 1320) and 4 in TB (100, 1550, 2370, and 2500).

## 3.4 Nucleotide substitution rate

Mean nucleotide substitution rates for TA and TB were estimated using both strict and relaxed uncorrelated molecular clocks, with 95% highest posterior density (HPD) intervals (Table 3). Substitution rates differed between the components: TA exhibited rates of $7.25 \times 10^{-4}$ (strict clock) and $6.22 \times 10^{-4}$ (relaxed clock), while TB showed higher rates of $8.15 \times 10^{-4}$ (strict clock) and $8.76 \times 10^{-4}$ (relaxed clock). Analysis of codon position (CoP) mutation rates revealed higher substitution rates at the first CoP in TA and at the second CoP in TB.

## 3.5 Genetic diversity indices

**3.5.1 GDIs in country-wise ToLCNDV populations.** Notably, all the inferred GDIs for TB were higher than TA, highlighting the highly dynamic nature of the TB population. This suggests greater genetic variability in TB, potentially driven by higher mutation rates, frequent recombination, or stronger selective pressures. These characteristics underscore TB's adaptability and capacity to rapidly respond to environmental changes or host-pathogen interactions (Table 4).

Country-specific analysis (Table 4) revealed the highest ToLCNDV genetic diversity in India, Pakistan, Spain, and Iran, and the lowest in Tunisia. Polymorphic sites (S) were most abundant in the TA-all (1777) and TB-all (1673) datasets, followed by India (TA: 1521; TB: 1466) and Pakistan (TA: 1027; TB: 831). Italy (TA: 57) and Tunisia (TB: 50) had the fewest. A similar trend was observed for the number of mutations (Eta [s]), with the TA-all (853.65) and TB-all datasets

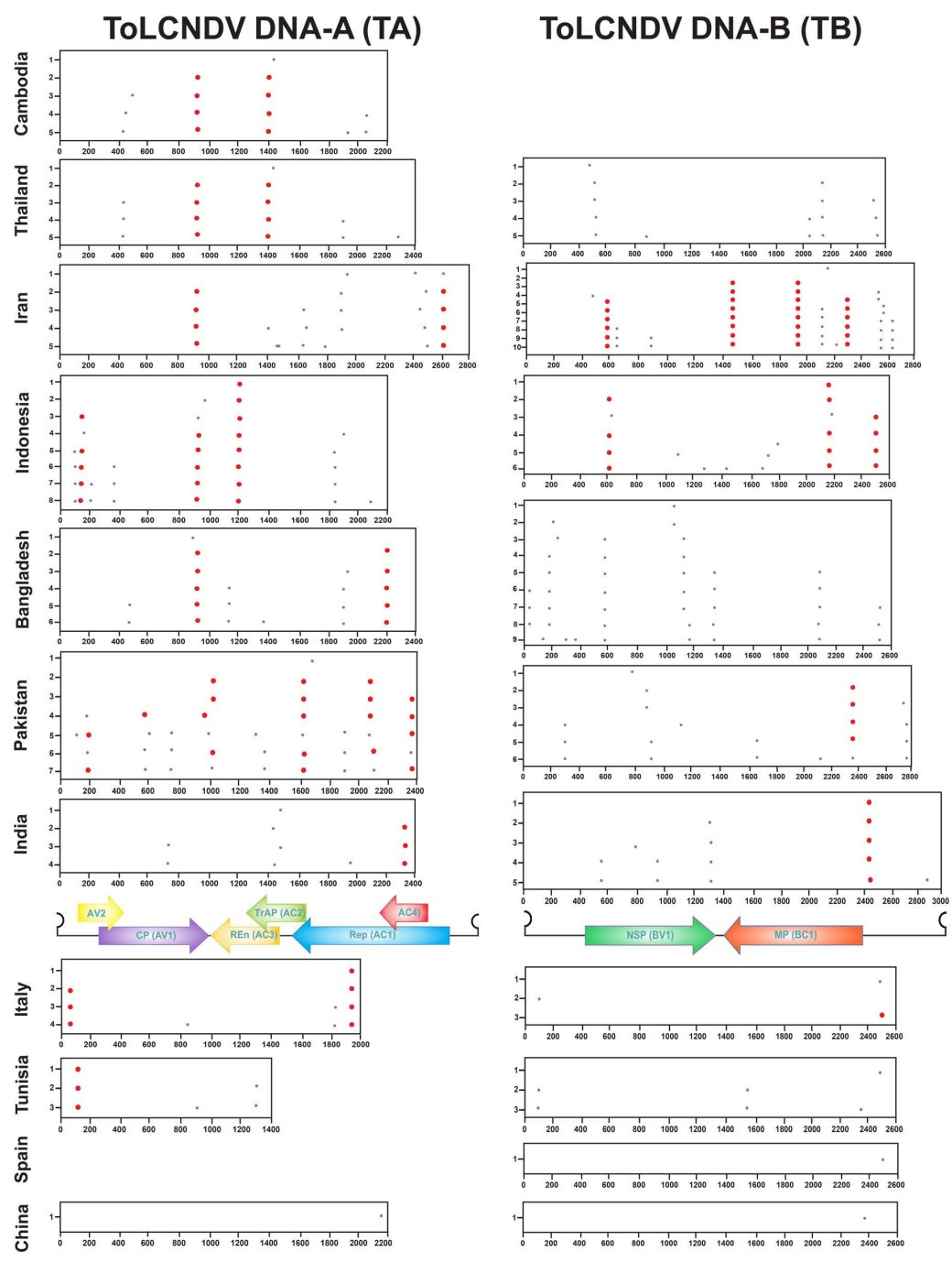

**Fig 4. Distribution of recombination breakpoints within TA and TB populations across different countries.** Well-supported breakpoints (high AICc scores) are shown as large red dots, while less supported breakpoints (lower AICc scores) are shown as smaller gray dots. The x-axis indicates nucleotide position (coordinates), and the y-axis represents the number of breakpoints. The linear genome organization of TA and TB is depicted between the plots.

**Table 3. Mean nucleotide substitution and codon position mutation rates of TA and TB.**

| | TA | | TB | |
|---|---|---|---|---|
| | **Strict clock (ESS value)** | **Relaxed clock (ESS value)** | **Strict clock (ESS value)** | **Relaxed clock (ESS value)** |
| Mean nt substitution rate (site$^{-1}$ year$^{-1}$) | $7.25 \times 10^{-4}$ (311) | $6.22 \times 10^{-4}$ (262) | $8.15 \times 10^{-4}$ (362) | $8.76 \times 10^{-4}$ (361) |
| At 95% HPD interval | $7.26 \times 10^{-4}$, $7.24 \times 10^{-4}$ | $6.23 \times 10^{-4}$, $6.21 \times 10^{-4}$ | $8.24 \times 10^{-4}$, $8.06 \times 10^{-4}$ | $9.68 \times 10^{-4}$, $8.44 \times 10^{-4}$ |
| CP1 mu | 1.084 (5927) | 1.195 (951) | 1.009 (5685) | 1.016 (2701) |
| CP2 mu | 0.926 (6094) | 0.974 (951) | 1.049 (7737) | 1.035 (675) |
| CP3 mu | 0.99 (4785) | 0.831 (951) | 0.942 (2577) | 0.949 (4854) |

**Table 4. Country-wise landscape of genetic diversity indices in ToLCNDV populations.**

| Country | Virus component | No. of seq | InDel sites | S | Eta (h) | Eta (s) | Hd | k |
|---|---|---|---|---|---|---|---|---|
| TA-all | TA | 662 | 221 | 1754 | 2886 | 853.65 | 0.927 | 142.99 |
| Bangladesh | | 12 | 3 | 480 | 553 | 323 | 1.00 | 152.93 |
| Cambodia | | 19 | 17 | 545 | 613 | 25 | 0.988 | 192.61 |
| China | | 27 | 0 | 104 | 107 | 60.67 | 0.994 | 15.09 |
| India | | 121 | 193 | 1521 | 2324 | 704 | 0.99 | 213.32 |
| Indonesia | | 3 | 0 | 224 | 229 | N/A | 1.00 | 151.0 |
| Iran | | 6 | 2 | 199 | 205 | 34.17 | 0.933 | 100.87 |
| Italy | | 5 | 0 | 57 | 58 | 21 | 1.00 | 30.50 |
| Laos | | 3 | 2 | 254 | 254 | N/A | 0.667 | 169.33 |
| Pakistan | | 352 | 198 | 1027 | 1334 | 721.94 | 0.748 | 25.27 |
| Spain | | 93 | 3 | 451 | 506 | 233.46 | 0.999 | 20.18 |
| Thailand | | 7 | 10 | 294 | 327 | 253 | 1.00 | 107.10 |
| Tunisia | | 8 | 14 | 157 | 45 | 208 | 1.00 | 13.96 |
| TB-all | TB | 535 | 223 | 1673 | 3035 | 1106.25 | 0.968 | 219.68 |
| Bangladesh | | 4 | 10 | 349 | 371 | 212 | 0.833 | 208.33 |
| India | | 94 | 185 | 1466 | 2528 | 785.55 | 0.998 | 309.99 |
| China | | 27 | 2 | 128 | 137 | 77.04 | 1.00 | 16.82 |
| Indonesia | | 3 | 20 | 239 | 250 | N/A | 1.00 | 163 |
| Iran | | 7 | 44 | 696 | 813 | 345.43 | 1.00 | 312.05 |
| Italy | | 5 | 2 | 76 | 76 | 29 | 1.00 | 39.80 |
| Pakistan | | 279 | 109 | 831 | 1404 | 350 | 0.863 | 114.10 |
| Spain | | 104 | 175 | 1048 | 1466 | 581 | 0.998 | 82.81 |
| Thailand | | 4 | 25 | 335 | 353 | 298 | 1.00 | 182.67 |
| Tunisia | | 8 | 0 | 50 | 50 | 31.50 | 0.964 | 15.39 |

N/A = program couldn't infer values for 3 or a smaller number of sequences.

Abbreviation used in the table are tomato leaf curl New Delhi virus DNA-A (TA), DNA-B (TB), insertions and deletions (InDels), total number of polymorphic (segregating) sites (S), total number of mutations [Eta (h)], total number of singleton mutations [Eta (S)], haplotype diversity (Hd), average number of nucleotide difference between sequences (k), nucleotide diversity ($\pi$), haplotypes (h), Watterson's estimate of the population mutation rate based on the total number of segregating sites ($\theta$w), Watterson's estimate of the population mutation rate based on the total number of mutations ($\theta$-Eta).

showing the highest values, followed by Pakistan (TA: 721.94), India (TA: 704), and Bangladesh (TA: 323). Italy (TA: 21) again had the fewest.

Insertions/deletions (InDels) were most frequent in the combined TA (221) and TB (223) datasets, followed by India (TA: 193; TB: 185) and Pakistan (TA: 198; TB: 109). No InDels were found in Italian or Indonesian TA isolates, or in Tunisian

TB isolates. Genetic differentiation (k) was highest in Indian TA isolates (213.32) and TB isolates (309.99), followed by Cambodia (TA: 192.61), Laos (TA: 169.33), Bangladesh (TA: 152.93) and Iran (TB: 312.05). Tunisia showed the lowest k values for both TA (13.96) and TB (15.39).

Polymorphism (θw) was highest in Indian TA isolates (0.109) compared to TA-all (0.099), Bangladesh (0.067), Laos (0.062), and Pakistan (0.061), with Tunisian TA isolates showing the lowest (0.005) (Fig 5A). For TB, TB-all had the highest θw (0.122), and Tunisia the lowest (0.0072) (Fig 5B). Nucleotide diversity (π) varied across datasets (Fig 5A and 5B). Among TA isolates, India (0.082), Cambodia (0.078), and Laos (0.062) showed the highest π, while Pakistan (0.0096) and Spain (0.0074) were lowest. For TB, India and Iran showed similarly high π, followed by TB-all (0.11) and Pakistan (0.089), with Tunisia having the lowest (0.0057). Overall, TA (average π < 0.07) had lower diversity than TB (average π ≈ 0.11).

Analysis of nucleotide diversity across ORFs (Fig 5C) showed decreasing π from NSP (0.11) to MP (0.078), REn (0.068), and Rep (0.058), with CP the lowest (0.044), suggesting strong purifying selection. Polymorphism (θw and θ–Eta) was highest in NSP, followed by AV2, Rep, and AC4, with MP the lowest, possibly due to functional constraints on viral movement.

Analysis of nucleotide diversity across all positions within the ORFs revealed distinct patterns of variation (Fig 6). Several regions exhibited higher-than-average diversity: the N-terminal (0.12) and middle (0.14) regions of NSP; the

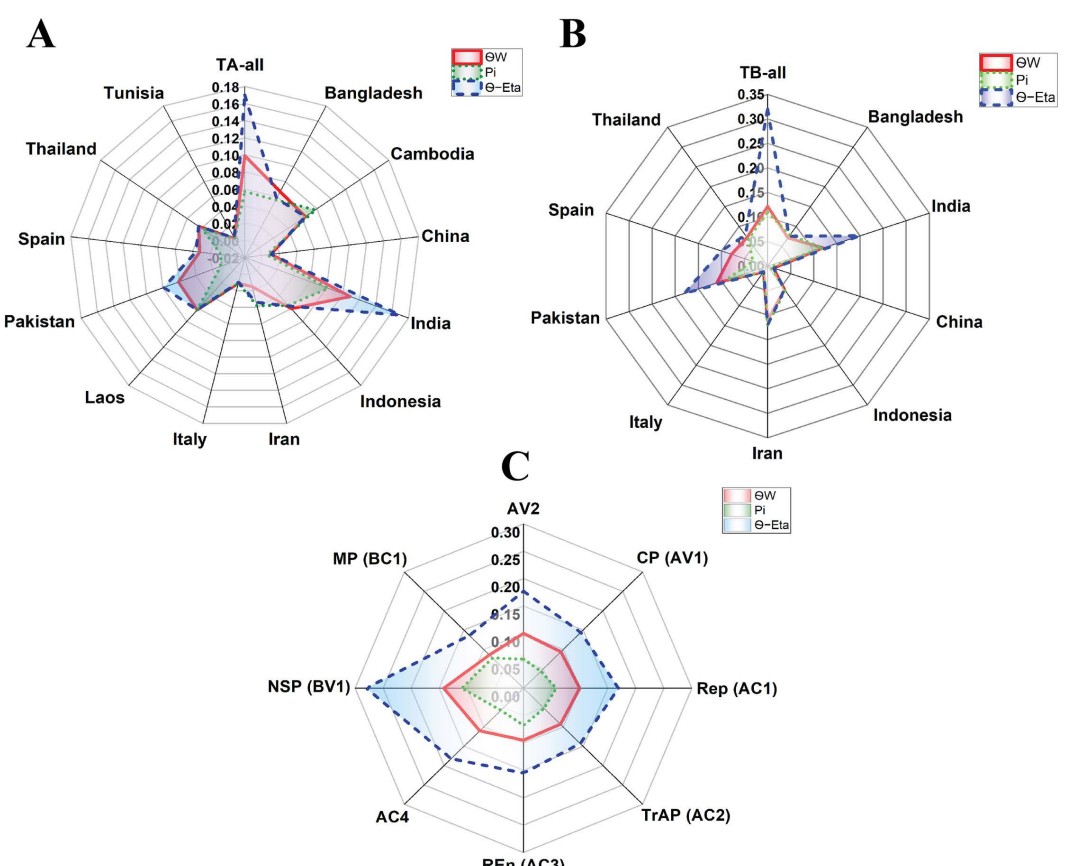

**Fig 5. The genetic diversity (π), Watterson's theta (θw), and theta index of the Eta (θ-Eta) estimated in populations of TA and its population in different countries (A), in TB (B), and different ORFs encoded by ToLCNDV (C).**

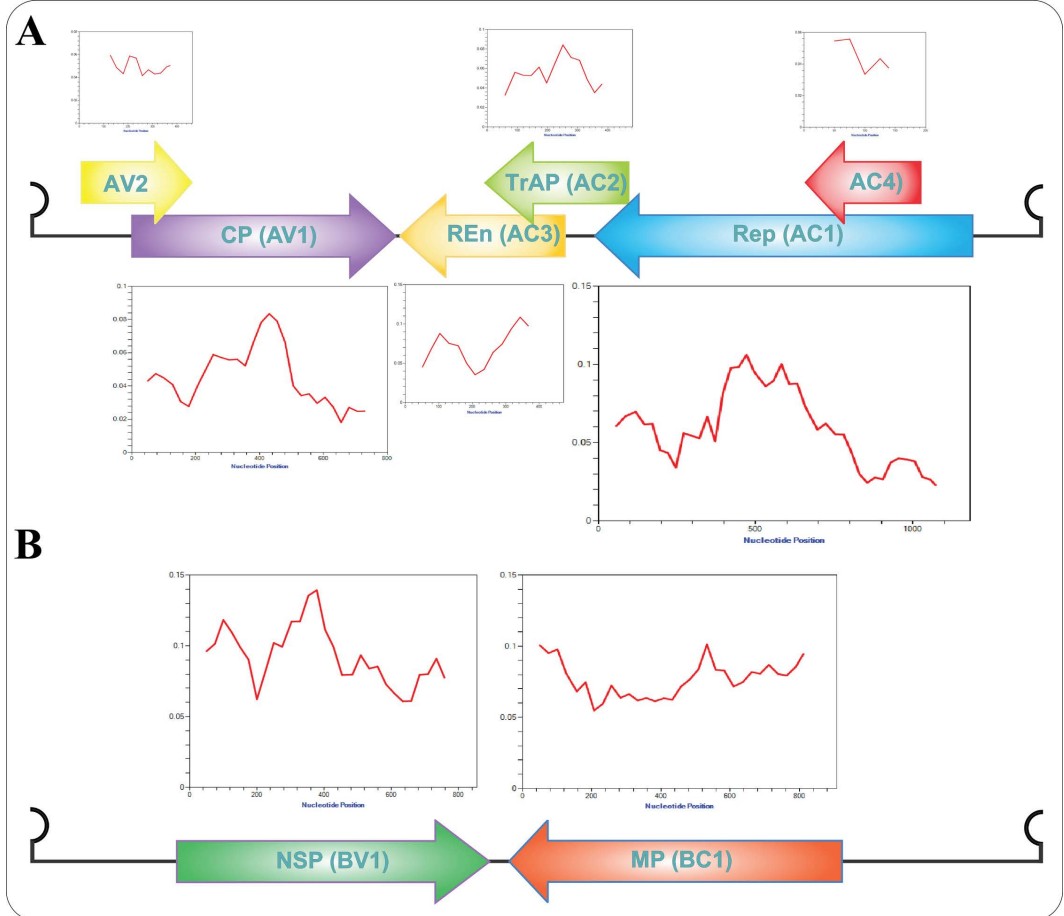

**Fig 6. The nucleotide diversity analysis across all the encoded ORFs in TA (A) and TB (B) populations.** The x-axes indicate the nucleotide positions, while the y-axes display the corresponding Pi values.

N-terminal, middle, and C-terminal regions of MP (0.1); the middle region of Rep (0.11); the N-terminal region of REn (0.11); and the middle regions of CP (0.084) and TrAP (0.082). Conversely, the regions where these ORFs overlap demonstrated lower levels of genetic diversity, suggesting evolutionary constraints and functional conservation within these shared sequences. This contrast highlights the selective pressures acting on different parts of the viral genome, with overlapping regions likely under stronger purifying selection to maintain functionality in multiple reading frames. Cross-country analysis (S2 Fig) showed similar diversity patterns across the genome for TA isolates from Pakistan, India, Bangladesh, Italy, and Spain, a trend largely mirrored in their corresponding TB isolates (excluding Italy). Conversely, TA isolates from Iran and Thailand exhibited divergent patterns but shared similarities within their respective groups, a trend also observed in their TB isolates, with Iranian isolates showing greater diversity.

**3.4.2 GDIs in ToLCNDV-encoded ORFs.** Analysis of GDIs in ToLCNDV-encoded ORFs (Table 4) revealed that within TA, Rep exhibited the highest values for most GDIs, while AV2 had the lowest. In TB, MP showed greater genetic diversity than NSP. CP was identified as the most diverse region based on S and Eta. MP showed the highest number of pairwise nucleotide differences (k), followed by Rep and CP, with NSP showing the least (Table 5).

Analysis of ORFs revealed contrasting levels of diversity. Within the S and Eta datasets, the coat protein (CP) region was the most diverse (S = 505, Eta = 286). Surprisingly, the smaller ORFs, AV2 and AC4, also exhibited considerable

**Table 5. Landscape of genetic diversity indices in ToLCNDV-encoded ORFs.**

| ORF | Virus component | No. of seq | InDel sites | S | Eta (h) | Eta (s) | Hd | k | h |
|---|---|---|---|---|---|---|---|---|---|
| AV2 | TA | 662 | 11 | 231 | 407 | 128 | 0.759 | 17.29 | 204 |
| CP | | 662 | 11 | 505 | 777 | 286 | 0.837 | 33.72 | 272 |
| Rep (AC1) | | 662 | 54 | 74 | 1252 | 286 | 0.859 | 61.60 | 316 |
| TrAP (AC2) | | 662 | 37 | 257 | 396 | 132 | 0.781 | 20.65 | 222 |
| REn (AC3) | | 662 | 41 | 253 | 415 | 127 | 0.783 | 26.01 | 224 |
| AC4 | | 662 | 32 | 246 | 415 | 98 | 0.792 | 18.61 | 214 |
| NSP (BV1) | TB | 535 | 25 | 67 | 132 | 34 | 0.883 | 7.56 | 111 |
| MP (BC1) | | 535 | 17 | 486 | 766 | 245 | 0.925 | 65.55 | 232 |

Abbreviation used in the table are tomato leaf curl New Delhi virus DNA-A (TA), DNA-B (TB), insertions and deletions (InDels), total number of polymorphic (segregating) sites (S), total number of mutations [Eta (h)], total number of singleton mutations [Eta (S)], haplotype diversity (Hd), average number of nucleotide difference between sequences (k), nucleotide diversity (π), haplotypes (h), coat protein (CP), replication associated protein (Rep), transcriptional activator protein (TrAP), replication enhancer protein (REn), movement protein (MP), nuclear shuttle protein (NSP), Watterson's estimate of the population mutation rate based on the total number of segregating sites (θw), and Watterson's estimate of the population mutation rate based on the total number of mutations (θ-Eta).

diversity (AV2: S = 231, Eta = 128; AC4: S = 246, Eta = 98), while NSP showed the lowest diversity (S = 67, Eta = 34). Insertion/deletion (InDel) analysis showed Rep with the highest number (54) and AV2 with the fewest (11). Overall, pairwise nucleotide differences (K) were highest for MP (65.55), followed by Rep (61.60) and CP (33.72), with NSP again showing the least divergence (7.56) (Table 5).

### 3.5 Neutrality indices

Neutrality tests (TD and FLD) revealed a significant influence of natural selection on ToLCNDV, with negative values observed across all datasets (Fig 7). These indices varied considerably by country. Spanish TA isolates exhibited the most negative TD (−2.63) and FLD (−3.04) values, followed by Pakistan (TD: −2.61; FLD: −7.34) and India (TD: −1.70; FLD: −1.81), indicating an excess of low-frequency polymorphisms and population expansion under purifying selection. Conversely, positive values in Bangladesh, Cambodia, Iran, and Italy suggested balanced selection and reduced population sizes due to low levels of both low- and high-frequency polymorphisms. Across all datasets, TD values were most negative in TA-all (−1.97), CP (−1.59), and TB-all (−1.57), while MP showed the least negative value (−0.28). Similarly, FLD values were most negative in CP (−6.58), TB-all (−5.93), and TrAP (−5.25), with NSP exhibiting the least negative value (−1.85).

### 3.6 SNPs analysis

SNP analysis revealed an uneven genomic distribution (Fig 1), with significantly higher SNP density in specific hotspots: the AV2 upstream region, the c-ter of CP, the M-region of Rep and TrAP. The TB population displayed greater sequence conservation than the TA population, particularly within coding regions compared to non-coding regions. These results suggest ToLCNDV population differentiation is influenced by demographic and evolutionary forces.

### 3.7 Estimation of selection pressure

Selection pressures on ToLCNDV ORFs were analyzed using dN/dS ratios, FUBAR, and SLAC methods (Table 6). dN/dS ratios generally indicated purifying selection across most ORFs. However, AC4 (2.17) and Rep (1.47) exhibited higher ratios, suggesting diversifying selection, while TrAP (0.28) and REn (0.31) showed the lowest ratios, indicating strong purifying selection. FUBAR analysis identified positively selected sites in all TA ORFs, with AV2, TrAP (AC2), and AC4

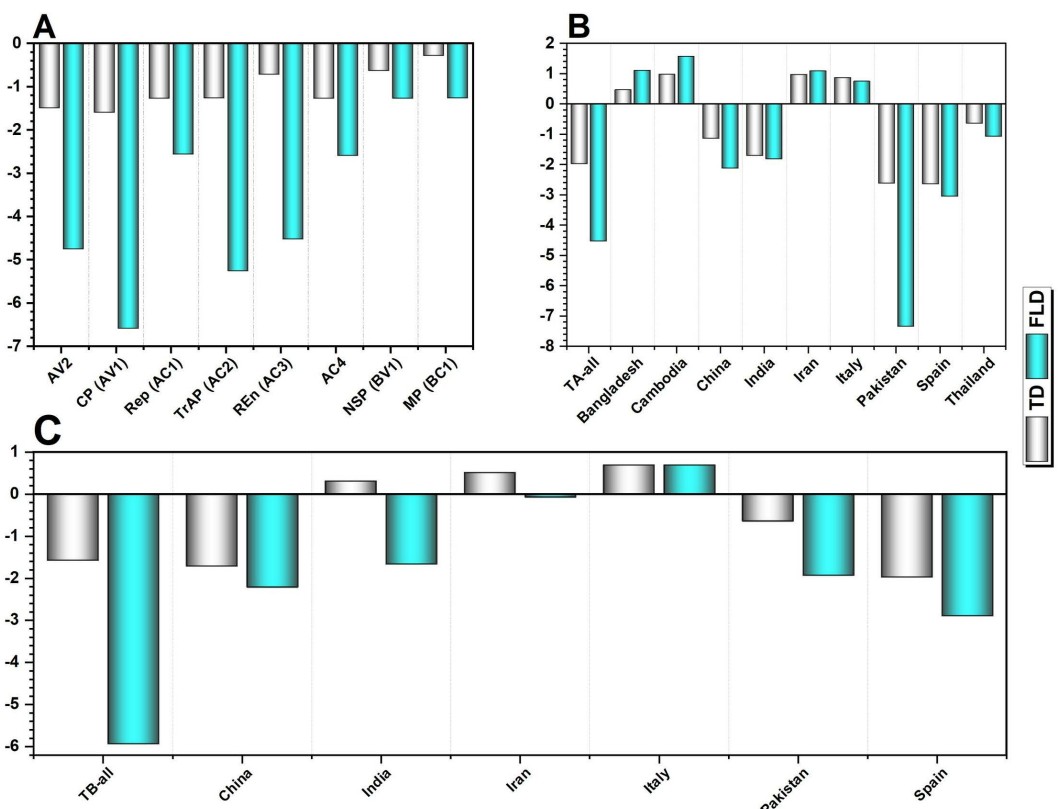

**Fig 7. Estimation of Tajima's D (TD) and Fu & Li's D (FLD) genetic diversity attributes in ToLCNDV-encoded ORFs (A), and its TA populations (B), and TB populations (C) in different countries.**

**Table 6. Sequence variability analysis of ORFs encoded by ToLCNDV.**

| Virus | ORF | Best Model | Mean distance (d) | dN | dS | dN/dS | Selection sites | | | |
|---|---|---|---|---|---|---|---|---|---|---|
| | | | | | | | FEL | | FUBAR | |
| | | | | | | | NS | PS | NS | PS |
| TA | AV2 | T92+G | 0.057±0.006 | 0.048±0.01 | 0.09±0.016 | 0.53 | 24 | 7 | 18 | 26 |
| | CP | TN93+G | 0.050±0.005 | 0.042±0.005 | 0.054±0.01 | 0.78 | 151 | 2 | 108 | 1 |
| | Rep (AC1) | HKY+G | 0.046±0.011 | 0.063±0.005 | 0.043±0.008 | 1.47 | 222 | 6 | 112 | 3 |
| | TrAP (AC2) | GTR+G | 0.049±0.008 | 0.036±0.006 | 0.129±0.024 | 0.28 | 29 | 13 | 11 | 13 |
| | REn (AC3) | GTR+G | 0.094±0.014 | 0.046±0.007 | 0.151±0.032 | 0.31 | 50 | 4 | 32 | 3 |
| | AC4 | TN93+G | 0.071±0.009 | 0.063±0.01 | 0.029±0.01 | 2.17 | 28 | 11 | 2 | 11 |
| TB | NSP (BV1) | T92+G | 0.11±0.008 | 0.092±0.008 | 0.118±0.017 | 0.78 | 143 | 3 | 78 | 0 |
| | MP (BC1) | HKY+G+I | 0.087±0.006 | 0.079±0.007 | 0.094±0.016 | 0.84 | 192 | 0 | 97 | 0 |

Abbreviation used in the table are tomato leaf curl New Delhi virus DNA-A (TA), DNA-B (TB), non-synonymous (dN), synonymous (dS), Tamura-Nei (TN93), Tamura 3-parameter (T92), jukes-cantor (JC), Kimura 2-parameter (K2), Gamma distribution (G), invariable (I),), coat protein (CP), replication associated protein (Rep), transcriptional activator protein (TrAP), replication enhancer protein (REn), movement protein (MP), nuclear shuttle protein (NSP), Tajima's D test (TD), Fu and Li's D test (FLD), positively selected sites (PS), negatively selected sites (NS), and not calculated due to just a few sequences (NC).

For the detection of positive selection, the GTR model was used with a statistical significance level ($p \leq 0.05$ for FEL and posterior probability $p \leq 1.5$ for FUBAR).

showing the most (26, 13, and 11, respectively), while Rep, REn, and CP had fewer. Conversely, Rep, CP, MP, and NSP exhibited more negatively selected sites. FEL analysis corroborated these trends, showing both positive and negative selection, but with substantially more sites under negative selection, particularly in Rep (222), MP (192), and CP (151).

## 4. Discussion

ToLCNDV is a significant pathogen, with exceptionally wide host and geographical range, infecting over 43 plant species [3,5–9,11–14], across Asia, Africa, and Europe. Its rapid spread and genetic adaptability have made it a critical subject of study in plant virology. This study provides a comprehensive analysis of the geographical distribution, phylogenetic relationships, recombination events, and genetic diversity of ToLCNDV, shedding light on its evolutionary dynamics and adaptation mechanisms.

ToLCNDV isolates harboured high variability and rapid evolution, corroborating to many RNA viruses [43,44]. This dynamic evolution, shaped by the virus's ecological niche, results in diverse genetic fingerprints and distinct regional groupings. The abundance of TA and TB sequences from South Asia, particularly Pakistan and India, identifies these regions as key diversity reservoirs, likely due to favorable agroecological conditions, localized selective pressures from host plants, vector (*B. tabaci*) populations, and environmental conditions. ToLCNDV's, combined with varying host resistance/susceptibility and differential transmission by specific whitefly cryptic species (e.g., Asia II-5 in Asia [32], Asia 1 [35], Mediterranean-Q1 [33], and MED-Q2 in Europe [34], further contributes to its complex evolutionary dynamics.

Phylogenetic analyses revealed significant regional structuring but limited host-specific segregation, with clear evolutionary connections between Indian and Spanish isolates. These relationships may reflect historical introductions and subsequent local adaptations, facilitated by trade and agricultural practices. This also suggests that geographical isolation and local adaptation play a more significant role in shaping ToLCNDV diversity than host specialization. The evolutionary time estimation pinpointing India as the origin of the most ancient ToLCNDV populations aligns with the region's long-standing agricultural history and diversity of host plants [52,65].The clustering of Indian and Spanish isolates (e.g., MT316381, MT316384, MT295294, KT175307) further supports the hypothesis of long-distance dispersal, possibly mediated by human activities such as trade or migration of insect vectors.

Recombination analysis using RDP and GARD identified extensive recombination events in both TA and TB components, with Asian isolates showing the highest frequency of recombination. Pakistani and Indian isolates dominated the recombination landscape, with 13 and 11 breakpoints in TA and 10 and 7 in TB, respectively. These findings align with the high genetic diversity observed in these regions, suggesting that recombination is a key driver of ToLCNDV evolution [47,49,66–68], consistent with the known propensity of begomoviruses to recombine during rolling circle replication [50,69]. Hotspot regions for recombination were identified in both TA (e.g., 100–200, 600–625, 1400–1425) and TB (e.g., 500–550, 1200–1300, 2400–2600), with interspecies recombination events indicating interactions between ToLCNDV and other begomoviruses. This trend, encompassing both inter- and intraspecies recombination, suggests location-dependent evolution, likely driven by factors such as host plants, intensive agriculture and mixed cropping, or environmental pressures in Asia that facilitate genetic exchange recombination [70]. The high recombination frequency in Asian isolates, particularly in Pakistan and India, contrasts with the limited recombination observed in European and African isolates, pointing to regional differences in evolutionary pressures and vector-host interactions [49,71]. Recombination is a key driver of viral evolution, facilitating rapid genetic exchange that impacts virulence, host adaptation, and resistance breakdown. In begomoviruses like ToLCNDV, recombination generates novel variants with enhanced fitness, enabling new host infections or resistance evasion [50]. Recombination in genes like CP or Rep can alter interactions with host defenses or vectors, increasing virulence and transmissibility [47,49]. Recombinant begomoviruses have been linked to resistance breakdown in crops by bypassing resistance genes [72]. Recombination also promotes host range expansion by incorporating genetic material from viruses adapted to different species [68,73].

The TB component of ToLCNDV exhibited significantly higher GDIs, segregating sites, SNPs, and overall genetic variation ($\pi$ and $\theta w$) than TA, underscoring its pivotal role in viral evolution [74,75]. While insertion/deletion (InDel) frequencies were comparable between the two components, TB exhibited greater sequence divergence, reflecting its dynamic evolutionary trajectory. Asian isolates displayed higher GDIs than their European and African counterparts, with Spanish TB isolates being an exception due to their elevated InDel and segregating site frequencies. Country-specific analysis revealed India as the most genetically diverse population for both TA and TB, followed by Pakistan and Spain. In contrast, Tunisia exhibited the lowest genetic diversity, possibly due to its isolated geographical location and limited agricultural trade. The high polymorphism ($\theta w$) and nucleotide diversity ($\pi$) in Indian and Pakistani isolates further underscore the role of these regions as hotspots for ToLCNDV evolution. At the ORF level, Rep showed the highest genetic variation, while AV2 exhibited the lowest, consistent with findings in other begomoviruses [46,49]. Specific regions within both TA and TB ORFs, particularly overlapping regions, exhibited reduced genetic diversity and SNP occurrence, suggesting stronger purifying selection to maintain functionality across multiple reading frames. This highlights the varying selective pressures acting on different parts of the viral genome. Moreover, this GDI pattern aligns with observations in other begomoviruses like pedilanthus leaf curl virus [49] and chili leaf curl virus (ChiLCV) [76], suggesting potential hotspots for mutation. In contrast, viruses like papaya leaf curl virus (PaLCuV) displayed low GDI but high rate of unique mutations [77], while tomato leaf curl Palampur virus (ToLCPalV) and euphorbia yellow mosaic virus showed minimal divergence and mutations [48,78]. This non-random distribution of genetic variation across the ToLCNDV genome, a characteristic feature of begomoviruses, suggests the influence of diverse factors on genetic variability at different genomic locations [76,77,79].

Selection pressure analysis (dN/dS, FUBAR, SLAC) revealed strong purifying selection across most ToLCNDV ORFs, but diversifying selection in AC4 (dN/dS = 2.17) and Rep (dN/dS = 1.47), reflecting distinct evolutionary pressures. AC4, a suppressor of RNA silencing, likely undergoes diversifying selection to evade evolving host defenses, as rapid adaptation to diverse host defense mechanisms drives amino acid changes [80]. Similarly, the Rep gene, critical for viral replication, faces diversifying selection due to its role in interacting with host replication machinery, necessitating variability to overcome host-specific barriers or optimize replication in new hosts [24]. Conversely, TrAP and REn showed strong functional constraints. Varying selection pressures and demographic histories are evident across regions. High neutrality indices in Pakistan, Spain, India, Thailand, and Tunisia suggest strong genetic conservation (possibly due to purifying selection) or recent population expansion [76,77,81,82]. Positive indices in Bangladesh, Cambodia, Iran, and Italy suggest a balance between purifying selection and smaller population sizes. The observed variable purifying selection, with stronger negative selection on essential genes (e.g., TrAP, REn) and diversifying selection on others (AC4, Rep), aligns with patterns in other viruses, including begomoviruses like ChiLCV, ToLCPalV, and PeLCV [48,76,83].

Mean nucleotide substitution rates (NSSY) were higher in TB ($8.15 \times 10^{-4}$ to $8.76 \times 10^{-4}$) compared to TA ($6.22 \times 10^{-4}$ to $7.25 \times 10^{-4}$), reflecting the greater genetic variability and adaptability of the TB component. This variability was further supported by higher genetic diversity indices (GDIs) in TB populations, particularly in India, Pakistan, and Iran (Table 4). This higher NSSY surpassed for several other geminiviruses, including ChiLCV, EACMV, PaLCuV, ToLCPalV, and TYLCV [45,48,49,76,77,84] and some RNA viruses [85], suggests a potentially rapid evolutionary rate for ToLCNDV. The NSSY also varied across different genomic regions, with AV2, Rep, REn, and AC4 genes showed higher rates than their counterparts in PeLCV [49]. While CP and TrAP genes showed lower NSSY, possibly due to their essential roles in virus structure and transmission. Notably, the choice of molecular clock model influenced NSSY estimates, with a strict clock being more suitable for TA and a relaxed clock for TB, a finding consistent with one previous study [77] but contrasting with another [49].

This study used stratified random sampling to ensure each country was represented proportionally to its population size, thus mitigating over- or under-representation. This approach inherently accounts for population size differences, with larger strata contributing more samples for statistical validity. While acknowledging the potential influence of unequal sample sizes on observed GDIs, isolates from smaller sample groups (like Cambodia, Indonesia, Tunisia, and Italy) still

exhibited higher GDIs than some larger groups, suggesting that observed differences are not solely due to sample size. Although limited sampling can mask some diversity, larger populations have more opportunities for mutation and recombination, driving increased diversity and potentially novel, more virulent variants. Therefore, observed GDI differences likely reflect a combination of sampling effects and genuine variations in regional viral population sizes.

This study elucidates the rapid evolutionary dynamics of ToLCNDV. The high genetic diversity, rapid nucleotide substitution rates, and frequent recombination events highlight the virus's adaptability. This adaptability poses significant challenges for resistance breeding and control measures as it can enable ToLCNDV to overcome existing resistance mechanisms. A multi-pronged approach, including continuous monitoring of viral populations, development of broad-spectrum resistance, implementation of integrated vector management (IVM) strategies, and exploration of novel control strategies, is necessary to manage ToLCNDV infections. IVM is critical to curb ToLCNDV spread, as whiteflies efficiently transmit the virus while developing insecticide resistance. IVM combines cultural controls (barrier crops, weed removal), biological agents (parasitoids like *Encarsia formosa*), and selective insecticides (neonicotinoid rotation) to disrupt transmission cycles and delay resistance evolution [86]. Understanding the influence of genetic, ecological, and geographical factors on ToLCNDV's evolution is crucial for developing effective control strategies against this dynamic virus.

## 5. Conclusions

This comprehensive analysis of ToLCNDV genetic diversity across three continents confirms its rapid evolution and high variability, typical of begomoviruses and comparable to some RNA viruses. Distinct regional structuring was observed, with South Asia (especially Pakistan and India) identified as a key diversity reservoir, likely due to favorable agroecological conditions, abundant hosts, and whitefly vector species. Notably, Asian isolates displayed significantly higher recombination rates, indicating location-dependent evolution. The TB component exhibited greater genetic diversity than TA, with ORF-specific variation. TB exhibited greater nucleotide diversity ($\pi = 0.11\%$) compared to TA ($\pi = 0.057\%$) and a slightly faster evolutionary rate ($8.15 \times 10^{-04}$ substitutions/site/year vs. $7.25 \times 10^{-04}$ for TA). Evidence of both purifying and diversifying selection was found across regions and ORFs. These findings emphasize the complex interplay of geography, host range, vector ecology, recombination, and selection in shaping ToLCNDV evolution and underscore the need for continued surveillance, region-specific management strategies to mitigate the emergence of more virulent variants, and balanced sampling in future diversity assessments.

## Supporting information

**S1 Fig. The distribution of recombination breakpoints within the global populations of TA (A) and TB (B).** Well-supported breakpoints, identified by high AICc scores, are highlighted as bold red dots. Conversely, breakpoints with lower AICc scores are represented by smaller grey dots. The linear organization of the genomes for TA and TB is also depicted.
(JPG)

**S2 Fig. Distribution of nucleotide diversity (π) in the genome of TA and TB in different countries.** The linear genome organization of TA and TB used to indicate the nt variation rate at each nt position and each ORF region. The window length was set 100 nt wide with a 25 nt step size.
(JPG)

**S1 Table. ToLCNDV sequences, their accession numbers, and details used in the study.**
(XLSX)

**S2 Table. Multiple recombination events detected by RDP in TA and TB genome components.**
(DOCX)

 

## Author contributions

**Conceptualization:** Zafar Iqbal.

**Data curation:** Zafar Iqbal.

**Formal analysis:** Zafar Iqbal, Sallah Ahmad Al Hashedi, Muhammad Naeem Sattar.

**Funding acquisition:** Adil Alshoaibi.

**Investigation:** Zafar Iqbal, Sallah Ahmad Al Hashedi.

**Methodology:** Zafar Iqbal, Sallah Ahmad Al Hashedi.

**Software:** Zafar Iqbal.

**Validation:** Zafar Iqbal.

**Writing – original draft:** Zafar Iqbal.

**Writing – review & editing:** Zafar Iqbal, Adil Alshoaibi, Sallah Ahmad Al Hashedi, Muhammad Naeem Sattar, Khaled Muhammad Amin Ramadan.

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
