## [Decision Letter · Decision Letter 0]

PONE-D-25-06105Deciphering the Genetic Landscape of ToLCNDV: Dynamic and Region-Specific Diversity Revealed by Comprehensive AnalysesPLOS ONE

Dear Dr. Iqbal,

Thank you for submitting your manuscript to PLOS ONE. After careful consideration, we feel that it has merit but does not fully meet PLOS ONE’s publication criteria as it currently stands. Therefore, we invite you to submit a revised version of the manuscript that addresses the points raised during the review process.

**Although the peer reviewers see merit in your work, they recommend a major revision. Therefore, the authors are requested to revise the manuscript by addressing all comments and responding to all criticisms raised by the reviewers.**

We look forward to receiving your revised manuscript.

Kind regards,

S.V. Ramesh, PhD

Academic Editor

PLOS ONE

**Journal Requirements:**

1. When submitting your revision, we need you to address these additional requirements. Please ensure that your manuscript meets PLOS ONE's style requirements, including those for file naming. The PLOS ONE style templates can be found at https://journals.plos.org/plosone/s/file?id=wjVg/PLOSOne_formatting_sample_main_body.pdf and https://journals.plos.org/plosone/s/file?id=ba62/PLOSOne_formatting_sample_title_authors_affiliations.pdf 2. Thank you for stating in your Funding Statement: This scientific paper is "derived from a research grant funded by the Research, Development, and Innovation " Authority (RDIA) - Kingdom of Saudi Arabia - with grant number (12877-KFU-2023-KFU-R-2-1-SE-). The publication of this research was supported by the Deanship of Scientific Research (DSR), King Faisal University, Kingdom of Saudi Arabia. Please provide an amended statement that declares *all* the funding or sources of support (whether external or internal to your organization) received during this study, as detailed online in our guide for authors at http://journals.plos.org/plosone/s/submit-now.  Please also include the statement “There was no additional external funding received for this study.” in your updated Funding Statement. Please include your amended Funding Statement within your cover letter. We will change the online submission form on your behalf. 3. Please note that your Data Availability Statement is currently missing the DOI/accession number of each dataset OR a direct link to access each database. If your manuscript is accepted for publication, you will be asked to provide these details on a very short timeline. We therefore suggest that you provide this information now, though we will not hold up the peer review process if you are unable. 4. We note that you have included the phrase “data not shown” in your manuscript. Unfortunately, this does not meet our data sharing requirements. PLOS does not permit references to inaccessible data. We require that authors provide all relevant data within the paper, Supporting Information files, or in an acceptable, public repository. Please add a citation to support this phrase or upload the data that corresponds with these findings to a stable repository (such as Figshare or Dryad) and provide and URLs, DOIs, or accession numbers that may be used to access these data. Or, if the data are not a core part of the research being presented in your study, we ask that you remove the phrase that refers to these data. 5. We note that Figure 2 in your submission contain map images which may be copyrighted. All PLOS content is published under the Creative Commons Attribution License (CC BY 4.0), which means that the manuscript, images, and Supporting Information files will be freely available online, and any third party is permitted to access, download, copy, distribute, and use these materials in any way, even commercially, with proper attribution. For these reasons, we cannot publish previously copyrighted maps or satellite images created using proprietary data, such as Google software (Google Maps, Street View, and Earth). For more information, see our copyright guidelines: http://journals.plos.org/plosone/s/licenses-and-copyright. We require you to either present written permission from the copyright holder to publish these figures specifically under the CC BY 4.0 license, or remove the figures from your submission: a. You may seek permission from the original copyright holder of Figure 2 to publish the content specifically under the CC BY 4.0 license.   We recommend that you contact the original copyright holder with the Content Permission Form (http://journals.plos.org/plosone/s/file?id=7c09/content-permission-form.pdf) and the following text:“I request permission for the open-access journal PLOS ONE to publish XXX under the Creative Commons Attribution License (CCAL) CC BY 4.0 (http://creativecommons.org/licenses/by/4.0/). Please be aware that this license allows unrestricted use and distribution, even commercially, by third parties. Please reply and provide explicit written permission to publish XXX under a CC BY license and complete the attached form.” Please upload the completed Content Permission Form or other proof of granted permissions as an "Other" file with your submission. In the figure caption of the copyrighted figure, please include the following text: “Reprinted from [ref] under a CC BY license, with permission from [name of publisher], original copyright [original copyright year].” b. If you are unable to obtain permission from the original copyright holder to publish these figures under the CC BY 4.0 license or if the copyright holder’s requirements are incompatible with the CC BY 4.0 license, please either i) remove the figure or ii) supply a replacement figure that complies with the CC BY 4.0 license. Please check copyright information on all replacement figures and update the figure caption with source information. If applicable, please specify in the figure caption text when a figure is similar but not identical to the original image and is therefore for illustrative purposes only.The following resources for replacing copyrighted map figures may be helpful: USGS National Map Viewer (public domain): http://viewer.nationalmap.gov/viewer/The Gateway to Astronaut Photography of Earth (public domain): http://eol.jsc.nasa.gov/sseop/clickmap/Maps at the CIA (public domain): https://www.cia.gov/library/publications/the-world-factbook/index.html and https://www.cia.gov/library/publications/cia-maps-publications/index.htmlNASA Earth Observatory (public domain): http://earthobservatory.nasa.gov/Landsat:
http://landsat.visibleearth.nasa.gov/USGS EROS (Earth Resources Observatory and Science (EROS) Center) (public domain): http://eros.usgs.gov/#Natural Earth (public domain): http://www.naturalearthdata.com/ 6. Please upload a new copy of Figure S1 as the detail is not clear. Please follow the link for more information: "https://blogs.plos.org/plos/2019/06/looking-good-tips-for-creating-your-plos-figures-graphics/" https://blogs.plos.org/plos/2019/06/looking-good-tips-for-creating-your-plos-figures-graphics/" 7. Please include captions for your Supporting Information files at the end of your manuscript, and update any in-text citations to match accordingly. Please see our Supporting Information guidelines for more information: http://journals.plos.org/plosone/s/supporting-information.

Reviewers' comments:

Reviewer's Responses to Questions

**Comments to the Author**

1. Is the manuscript technically sound, and do the data support the conclusions?

Reviewer #1: No

Reviewer #2: Yes

Reviewer #3: Yes

Reviewer #4: Yes

2. Has the statistical analysis been performed appropriately and rigorously? 

Reviewer #1: N/A

Reviewer #2: Yes

Reviewer #3: Yes

Reviewer #4: Yes

3. Have the authors made all data underlying the findings in their manuscript fully available?

Reviewer #1: No

Reviewer #2: Yes

Reviewer #3: Yes

Reviewer #4: Yes

4. Is the manuscript presented in an intelligible fashion and written in standard English?

Reviewer #1: No

Reviewer #2: Yes

Reviewer #3: Yes

Reviewer #4: Yes

5. Review Comments to the Author

**Reviewer #1:**  Dear Author

This MS was made based on totally online data available in GenBank, no original work.

Why this work was not done by using ToLCV sequences submitted from KSA?

All the data is used from GenBank and nothing new data in this MS, all are old.

In the introduction part, authors have mentioned that many countries but no KSA? Why?

In materials and methods section: From these datasets, sub-datasets were generated for each open reading frame (ORF). Additionally, sequences were categorized by country of origin, with further division into their respective ORFs to facilitate country-specific analyses. Where is KSA?

Results section: The figures and Tables: No data from KSA at all

Some figures are not with high resolution and should be replaced with high resolution fig.

The phylogenetic tree images are not at all visible.

Authors should provide the genetic diversity/similarity % in a table.

Figure 3. The phylogenetic dendrogram of selected sequences of ToLCNDV, TA (A) and TB (B), based on their respective full-length sequences. The phylogenetic tree was constructed with Maximum-Likelihood (ML) algorithm in MEGA11 with 1000 bootstrap values. Geographic origin (country) of the isolates is indicated by the coloured circle. Why not some sequences from KSA was used in this study?

RDP generates some important figures and data and this MS lacking those images.

Recombination breakpoints (start and end points) in ToLCNDV genome should be reflected in the image.

All the figures and Data is made and generated by softwares, there is no novel findings in this MS and does not reflect the originality of the work.

The accession numbers, name of the viruses’ acronyms are not available in this MS.

Discussion: This study used stratified random sampling to ensure each country was represented Proportionally to its population size, thus mitigating over- or under-representation.

Where is KSA data? If the funding agency from KSA then KSA data should be used so that the generated information will be valuable for KSA.

The generated data from this work is not as much as useful for scientific community as well as for KSA.

There are some grammatical errors were observed in the text.

Conclusion: Notably, Asian isolates displayed significantly, higher recombination rates, indicating location-dependent evolution. Again, KSA data missing.

Thanks

**Reviewer #2:**  I have critically reviewed the manuscript title “Deciphering the Genetic Landscape of ToLCNDV: Dynamic and Region-Specific Diversity Revealed by Comprehensive Analyses”, which describes the genetic diversity and evolutionary tracks of the most important tomato infecting bipartite begomovirus ToLCNDV. The manuscript is about very important issue of devastating begomoviruses which belong the largest genus of plant infecting viruses with wide host range. The manuscript utilized the advanced bioinformatics softwares to analyze the nucleotide mutation rate and recombination events in more than 650 virus complete genomes. The manuscript is well written describing the importance of study supported with update review of literature. Methodology of experiments are well explained to reproduce the experimental results. The results and discussions are comprehensively described. Although there are few typo errors which can corrected by careful reading of the manuscript.

I would recommend the acceptance of manuscript for publication in its present form.

**Reviewer #3:**  This paper studied “Deciphering the Genetic Landscape of ToLCNDV: Dynamic and Region-Specific Diversity Revealed by Comprehensive Analyses”. In general, this is a well written paper and the results support the main conclusion. Here, I would like to give few suggestions to further improve the manuscript:

1, The abstract can be more precise.

2, Briefly discuss how recombination may impact virulence, host adaptation, or resistance breakdown.

3, Briefly discuss why AC4 and Rep show diversifying selection while other genes are under purifying selection.

4, Emphasize the importance of integrated vector management (IVM) to limit viral spread.

**Reviewer #4:**  ToLCNDV is one of the important viral pathogens worldwide. Selection of problem is good. Manuscript is written nicely.

I have gone through the manuscript, the following corrections were raised and few highlighted in manuscript.

Quality of images of Fig 3 and Fig 4 should be improved.

Include all crops also and their distribution across the continents.

Include association of satellite molecules.

6. PLOS authors have the option to publish the peer review history of their article (what does this mean? ). If published, this will include your full peer review and any attached files.

**Do you want your identity to be public for this peer review?** For information about this choice, including consent withdrawal, please see our Privacy Policy .

Reviewer #1: No

Reviewer #2: **Yes: ** Dr. Khadim Hussain

Reviewer #3: No

Reviewer #4: No

---

## [Author Response · Author response to Decision Letter 1]

8 May 2025

All the comments and suggestions by the reviewers have been incorporated in the revised script. Please see the attached "Response to journal and Reviewers" document.

---

## [Decision Letter · Decision Letter 1]

Deciphering the genetic landscape of tomato leaf curl New Delhi virus: Dynamic and region-specific diversity revealed by comprehensive sequence analyses

PONE-D-25-06105R1

Dear Dr. Iqbal,

We’re pleased to inform you that your manuscript has been judged scientifically suitable for publication and will be formally accepted for publication once it meets all outstanding technical requirements.

Kind regards,

S.V. Ramesh, PhD

Academic Editor

PLOS ONE

Additional Editor Comments (optional):

Reviewers' comments:

Reviewer's Responses to Questions

**Comments to the Author**

1. If the authors have adequately addressed your comments raised in a previous round of review and you feel that this manuscript is now acceptable for publication, you may indicate that here to bypass the “Comments to the Author” section, enter your conflict of interest statement in the “Confidential to Editor” section, and submit your "Accept" recommendation.

Reviewer #5: All comments have been addressed

2. Is the manuscript technically sound, and do the data support the conclusions?

Reviewer #5: Yes

3. Has the statistical analysis been performed appropriately and rigorously? 

Reviewer #5: Yes

4. Have the authors made all data underlying the findings in their manuscript fully available?

Reviewer #5: Yes

5. Is the manuscript presented in an intelligible fashion and written in standard English?

Reviewer #5: Yes

6. Review Comments to the Author

Reviewer #5: (No Response)

7. PLOS authors have the option to publish the peer review history of their article (what does this mean? ). If published, this will include your full peer review and any attached files.

**Do you want your identity to be public for this peer review?** For information about this choice, including consent withdrawal, please see our Privacy Policy .

Reviewer #5: No

---

## [Editor Report · Acceptance letter]

PONE-D-25-06105R1

PLOS ONE

Dear Dr. Iqbal,

I'm pleased to inform you that your manuscript has been deemed suitable for publication in PLOS ONE. Congratulations! Your manuscript is now being handed over to our production team.

Kind regards,

on behalf of

Dr. Shunmugiah Veluchamy Ramesh

Academic Editor

PLOS ONE